# Position: Stop Overvaluing Multi-Agent Debate—We Must Rethink Evaluation and Embrace Model Heterogeneity

## Abstract

Multi-agent debate (MAD) has gained significant attention as a promising line of research to improve the factual accuracy and reasoning capabilities of large language models (LLMs). Despite its conceptual appeal, current MAD research suffers from critical limitations in evaluation practices, including limited benchmark coverage, weak baseline comparisons, and inconsistent setups. This paper presents a systematic evaluation of 5 representative MAD methods across 9 benchmarks using 4 foundational models. Surprisingly, our findings reveal that MAD often fail to outperform simple single-agent baselines such as Chain-of-Thought and Self-Consistency, even when consuming significantly more inference-time computation. To advance MAD research, we further explore the role of model heterogeneity and find it as a universal antidote to consistently improve current MAD frameworks. Based on our findings, we argue that the field must stop overvaluing MAD in its current form; for true advancement, we must critically rethink evaluation paradigms and actively embrace model heterogeneity as a core design principle.

## 1 Introduction

The age-old saying, *two heads are better than one*, encapsulates the enduring lesson that collaboration often triumphs over solitary effort. Can this human wisdom be applied to enhance the capabilities of large language models (LLMs)? An emerging line of research—commonly known as multi-agent debate or discussion (MAD)—suggests it can. Research has shown that after multiple LLM agents independently produce initial answers to a question, by having them engage in several rounds of discussing and reviewing answers from each other, they can improve the factual accuracy and reasoning quality of their final aggregated response [1]. As such, LLM performance is enhanced at inference time, without the need for additional training, suggesting MAD as an inference-time solution to boosting LLM capabilities. Thus, unsurprisingly, this line of research has garnered significant attention, with prestigious venues witnessing a surge in the number of publications [2, 3, 4, 5, 6, 7, 8].

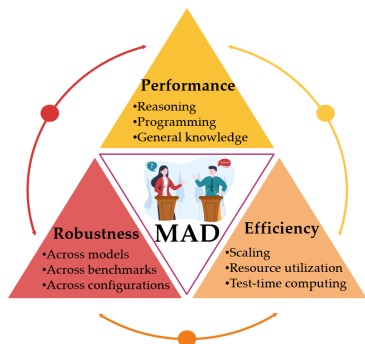

Figure 1: The promotion and recognition of MAD research require a systematic and comprehensive evaluation.

However, we find that this field has thus far suffered from a glaring lack of sufficiently systematic and comprehensive evaluations. The datasets used for evaluation are limited and have minimal overlap between different MAD methods (see Appendix A for details). While some focus on mathematical reasoning [1, 2], others may target machine translation [3], or

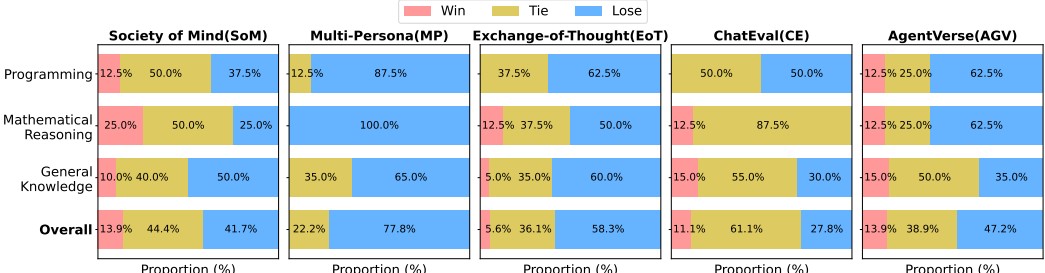

Figure 2: Performance comparison of MAD across 4 LLMs and 9 benchmarks, covering 3 top-level categories: general knowledge, mathematical reasoning, and programming. Each bar represents the distribution of conditions where MAD is better/comparable/worse than CoT. We employ the ANOVA test to evaluate differences among group means, considering a $p$-value greater than 0.05 as an indicator that no significant differences—a tie—exist.

programming tasks [4]. In some cases, a newly proposed MAD method is evaluated on another new dataset introduced in the same paper, which has not been fully disclosed [5]. Moreover, newly proposed methods are often compared solely against the very basic approach of directly prompting LLM to generate answers. This overlooks simple single-agent techniques, such as Chain-of-Thought [9], as well as more recent advancements in MAD. In addition, while evaluations are frequently performed using only proprietary LLMs, the efficiency of MAD that trade-off token consumption and performance gains have rarely been considered, with only few exceptions [7].

These current common practices of MAD evaluation can easily give rise to significant doubts about the reproducibility and generalizability of MAD methods, while also creating confusion about the status quo of MAD research. Specifically, do MAD methods genuinely outperform existing methods? To what extent do they outperform simpler single-agent baselines? Among them, what is the state of the art? Do these methods demonstrate robust performance across diverse conditions, or is their success a mere result of cherry-picking—critically dependent on specific dataset choices and parameter settings? Do they excel in both performance and efficiency? Undoubtedly, as long as these questions remain unresolved, they will continue to hinder the recognition and promotion of MAD research, even eroding the trust in MAD research, just as they have plagued many other areas of ML [10].

To this end, in this paper, we critically evaluate 5 representative MAD methods across 9 widely adopted benchmarks, spanning various key high-level LLM capabilities: general knowledge, mathematical reasoning, and programming, with 4 different LLMs and various parameter configurations. As illustrated in Figure 1, our evaluation is structured around three pivotal dimensions: performance, efficiency, and robustness. Our first set of experiments, comparing these MAD methods to single-agent approaches, including Chain-of-Thought (CoT) and Self-consistency (SC) [11], reveal an astonishing negative result—*these MAD methods generally fail to outperform CoT*, despite CoT being much simpler. As summarized in Figure 1, none of these MAD methods achieve a win rate higher than 20% when compared to CoT across 36 scenarios (4 models × 9 benchmarks). Further experiments varying the number of agents and debate rounds show that simply adjusting these hyperparameters can hardly reverse this negative outcome. Moreover, the underperformance becomes even more pronounced when compared to SC, especially when using a comparable number of LLM calls or tokens.

All these results point to a crucial reality—*existing MAD approaches are less effective than currently believed, even underperforming simple single-agent methods like CoT and SC*. Such a conclusion would remain obscured under today's fragmented and selective evaluation norms. AS detailed in Table 3, prior MAD research share minimal overlap in benchmarks. This brings us to the first half of our position: **we must stop overvaluing multi-agent debate and rethink how we evaluate MAD research.** The persistent overvaluation of MAD methods, fueled by selective benchmarks and inconsistent baselines, has distorted our understanding of their true utility. To advance, the community must adopt a more rigorous, transparent, systematic, and comprehensive evaluation paradigm—one that includes consistent comparisons with strong single-agent baselines, diversified and representative benchmark suites, and a principled analysis of the trade-offs between performance gains and computational cost.

While the above results seem discouraging, we emphasize that they do *not* imply MAD research is a frustrating dead end. After all, countless examples have demonstrated the power of human collaboration. Then, we must ask: do current MAD methods truly emulate how people engage in productive discussion? A key factor enabling meaningful discussion is the diversity of knowledge and reasoning among individuals. While different LLMs trained on different data and paradigms may exhibit distinct strengths likewise, this crucial aspect remains largely unexplored in current MAD research—where, when multiple LLM agents are employed, they are typically instantiated from the same model. To allow for a broad validation on this position, we introduce a simple twist that can be incorporated into *any* existing MAD methods:[1] instead of relying on a single model, agents randomly select an LLM from a diverse set of candidate models at inference time to generate responses. Despite its simplicity, this twist proves to be broadly effective, leading to performance gains for *all* MAD methods considered.

Thus, the second half of our position is clear—**we must embrace model heterogeneity as a foundational principle in MAD research**. Using identical agents from the same model undermines the very premise of debate—diverse knowledge and reasoning. By simply drawing agents from a pool of distinct LLMs, we introduce real epistemic diversity, leading to consistently better outcomes. This highlights that progress in this field depends not only on new algorithms or clever agent debating strategies, but also on a foundational rethinking of what it means to debate.

This position paper is structured as follows. Section 2 briefly discusses the landscape of MAD research. Section 3 presents a comprehensive evaluation of representative methods. Section 4 explores the potential of model heterogeneity. Section 5 proposes multiple key research questions for future work based on our analysis. Section 6 concludes the paper.

## 2 Background

MAD is one of the frontiers of LLM research due to their potential to enhance the reasoning and decision-making capabilities of LLMs. At their core, MAD methods share several common principles. In the seminal work **SoM** [1]—commonly referred to as MAD in most subsequent studies—this approach involves multiple agents following three steps to generate the final response: (1) Response Generation, where each agent produces an initial solution based on its unique perspective; (2) Debate, where agents debate to identify logical inconsistencies or knowledge gaps; and (3) Consensus Building, where the consensus is determined by majority voting or a judge agent.

Subsequently, numerous studies improve MAD from different perspectives: (1) A series of following works explored enhancing the reasoning capabilities of MAD by assigning different roles to agents [12, 3, 2, 13]. (2) A part of the research focused on improving communication topology [14, 15, 4, 16]. (3) Another line of work enhanced the way to exchange and integrate information between agents [17, 18, 5].A more detailed discussion on prior MAD research is provided in Appendix A.1.

To illustrate the landscape of MAD research, we briefly describe a selection of representative approaches. **Multi-Persona (MP)** [3] incorporates an affirmative agent (angel) and a negative agent (devil) presenting their answer to a judge agent, which ultimately determines the final solution. **Exchange-of-Thoughts (EoT)** [2] assigns three diverse roles to agents: detail-oriented nature, diligence, and problem-solving abilities. Additionally, it implements a confidence evaluation mechanism designed to reduce the adverse effects of erroneous reasoning processes. In **AgentVerse** [4], the verifier can dynamically determine the subsequent execution of MAD processing, allowing dynamic adjustment of communication topology. **ChatEval** [5] explores communication strategies among agents through three frameworks, focusing on the impact of asynchronous responses and round-by-round summarization on agent performance.

Each of the aforementioned methods has been published in prestigious venues and has garnered significant attention, with some accumulating hundreds of citations. However, despite their prominence, we observe notable shortcomings in their evaluation practices. As shown in Table 3 in Appendix A.1, the selected benchmarks vary widely across studies, with minimal overlap. In many cases, the benchmarks used to evaluate one MAD framework are entirely absent from the evaluations of others.

---

[1]We intentionally refrain from delving into well-calibrated technical solutions, such as designing a MAD framework explicitly optimized for leveraging model heterogeneity, as we believe that the development of these solutions holds significant promise, yet fall outside the scope of position papers. Instead, these solutions merit further exploration within the community.

Moreover, the scope of benchmark coverage is often limited. For example, FORD evaluates only on commonsense reasoning tasks, and CoMM is assessed solely on MMLU. MP considers just two datasets, one of which is a newly introduced dataset that has yet been fully disclosed. Last but not least, we also observe a lack in sufficient comparison with baselines. None of aforementioned MAD research include direct comparison with one another, which hinders a comprehensive understanding of the relative strengths of different approaches. Moreover, 4 of 7 methods[2] only compare against a single agent's direct response, overlooking simple single-agent baselines like CoT prompting simply instructing "Let's think step by step".

These fragmented and inconsistent evaluation practices obscure our understanding of the true utility and generalizability of MAD methods. This raises concerns about whether the reported strong performance is tied to specific benchmark choices and may not generalize to broader settings. Notably, the study [7] poses the critical question, "Should we be going MAD?" in the paper title and reports that MAD methods do not reliably outperform other SC. Similarly, the study [6] also critically asks "Are Multi-Agent Discussions the Key?" and reports that single-agent baselines with deliberately designed prompts (task-specific demonstrations) can match or even surpass the best existing MAD methods. However, the study [7] specifically focuses on medical prompting methods and medical benchmarks. In contrast, the study [6] covers a range of reasoning tasks though, providing task-specific demonstrations to single-agent baselines could constitute an unfair comparison with MAD methods. Therefore, these studies, while raising important concerns, still fall short of delivering conclusive answers.

# 3 Revisiting the Status Quo: A Comprehensive Evaluation

In light of the current landscape of MAD research, we argue that we must rethink how we evaluate these methods. To substantiate this claim, we conduct a comprehensive evaluation encompassing 5 representative MAD methods, evaluated across 9 widely adopted benchmarks and 4 foundation models. This extensive experimental setup is intentionally designed to uncover critical blind spots in the literature—blind spots that arise from current evaluation limitations.

## 3.1 Experimental Setup

**Datasets** We conduct our evaluation on 9 widely adopted standard benchmarks that cover 3 top-level capabilities of LLMs: general knowledge, mathematical reasoning, and programming. The benchmarks are: MMLU [19], MMLU-Pro [20], AGIEval [21], CommensenseQA [22], ARC-Challenge [23], GSM8K [24], MATH [25], HumanEval [26], and MBPP [27]. We briefly summarize their basic information in Table 4 in Appendix C.1, where more details are included.

**Foundation models** We consider both proprietary LLMs and open-sourced LLMs. They are *gpt-4o-mini-2024-07-18*, *claude-3-5-haiku-2024-1022*, *Llama3.1:8b-instruct*, and *Llama3.1:70b-instruct*. Unless stated otherwise, we maintain consistent inference configurations throughout our evaluation, setting the temperature $T = 1$ and $top\text{-}p = 1$, to balance generation quality and diversity.

**MAD methods and baselines** We consider five representative MAD frameworks and three single-agent baselines: single-agent (SA), Chain-of-Thought (CoT) [9], Self-Consistency (SC) [11], Society-of-Minds (SoM) [1], Multi-Persona (MP) [3], Exchange-of-Thoughts (EoT) [2], AgentVerse [4], and ChatEval [5]. SA simply prompts the agent with only the necessary problem description to generate the response. CoT prompts the agent with "Let's think step by step" to elicit step-by-step reasoning. SC repetitively samples from a CoT agent and utilizes majority voting to determine the final answer. SoM is the first MAD method proposed, serving as the foundation of a number of recent attempts [3, 14, 17, 16]. MP, EoT, AgentVerse, and ChatEval, are representative MAD frameworks that differ in their approaches to role-play, communication, answer aggregation, as summarized in Table 5 in Appendix C.2. Despite these differences, they have all attracted significant interest.

For all MAD methods considered,we follow the authors' open-source implementations. For fair comparison across different MAD methods, we slightly adjust the number of debate rounds of these methods to ensure that they all align to a similar amount of inference budget measured by the number of LLM calls. Unless otherwise mentioned, we consider the number of LLM calls to be 6, following the convention [1, 5]. We present more implementation details, including agents'

---

[2]SoM, ChatEval, MP, FORD lacked a comprehensive comparison against CoT.

Table 1: Performance results on GPT-4o-mini. We use lightred / lightblue to denote results higher/lower than CoT.

| Dataset | MMLU | MMLU-Pro | CommensenseQA | ARC-Challenge | AGIEval | GSM8K | MATH | HumanEval | MBPP |
|---|---|---|---|---|---|---|---|---|---|
| SA | 65.33 ± 0.93 | 58.07 ± 0.50 | 79.47 ± 0.25 | 88.27 ± 0.41 | 63.87 ± 1.05 | 91.13 ± 0.34 | 71.67 ± 1.31 | 66.67 ± 1.15 | 58.11 ± 0.66 |
| CoT | 80.73 ± 0.34 | 62.80 ± 0.99 | 82.87 ± 0.25 | 93.53 ± 0.41 | 66.40 ± 1.30 | 93.60 ± 0.82 | 72.87 ± 1.20 | 78.05 ± 1.49 | 62.26 ± 0.84 |
| SC | 82.13 ± 0.66 | 66.27 ± 1.39 | 83.80 ± 0.28 | 93.93 ± 0.25 | 67.07 ± 0.84 | 95.67 ± 0.19 | 73.96 ± 0.54 | – | – |
| SoM | 74.73 ± 0.52 | 62.80 ± 1.02 | 80.73 ± 0.93 | 90.80 ± 0.43 | 64.33 ± 0.34 | 94.93 ± 0.34 | 75.40 ± 0.71 | 68.09 ± 1.25 | 56.94 ± 1.12 |
| MP | 75.47 ± 0.84 | 60.53 ± 1.27 | 68.07 ± 1.57 | 90.27 ± 0.25 | 61.67 ± 1.43 | 90.87 ± 0.19 | 51.87 ± 0.66 | 63.01 ± 2.30 | 45.78 ± 0.80 |
| EoT | 67.87 ± 0.41 | 61.20 ± 0.65 | 80.07 ± 0.52 | 86.40 ± 0.28 | 65.07 ± 0.66 | 91.40 ± 0.57 | 75.93 ± 1.23 | 73.78 ± 2.17 | 56.16 ± 0.49 |
| ChatEval | 79.13 ± 0.90 | 62.20 ± 0.49 | 81.07 ± 0.84 | 93.20 ± 0.28 | 68.87 ± 0.94 | 93.60 ± 0.00 | 69.36 ± 1.58 | 71.75 ± 0.76 | 53.70 ± 0.55 |
| AgentVerse | 80.40 ± 0.00 | 62.07 ± 0.52 | 80.73 ± 0.41 | 92.47 ± 0.09 | 63.87 ± 1.23 | 92.73 ± 0.50 | 64.49 ± 1.38 | 85.57 ± 1.25 | 58.88 ± 0.18 |

prompts, communication strategies, and agent roles in Appendix C.2. Our code is publicly available at https://anonymous.4open.science/r/MAD-eval-E4C4/ for reproducibility.

## 3.2 Experimental results

**Does MAD outperform simple single-agent baselines?** We first compare MAD frameworks to single-agent baselines to assess their relative performance. For robustness, we repeated the experiments three times, reporting the standard deviations. Table 1 and Tables 6, 7, 8 in Appendix D present empirical results on GPT-4o-mini, Llama3.1-8b, Llama3.1-70b, Claude-3.5-haiku, respectively. In these tables, all methods are compared against CoT, with results higher or lower than CoT denoted in lightred and lightblue , respectively. Our results indicate that MAD methods fail to consistently outperform CoT across different models and benchmarks. Specifically, in Table 1, SoM underperforms CoT on all datasets utilizing the GPT-4o-mini model. Similarly, more advanced frameworks such as ChatEval and AgentVerse merely outperform CoT on one out of nine datasets. Furthermore, analyses across other models reveal that while MAD frameworks occasionally achieve better performance, they generally underperform CoT. For instance, AgentVerse is the only MAD framework that outperforms CoT on MMLU using Claude-3.5-haiku, achieving a +0.85% performance gain. However, all other MAD frameworks underperform CoT by at least -3.60%. When being compared to SC, the underperformance of MAD approaches is more noticeable. In most cases when SC can be applied[3], SC achieves the highest performance, defeating CoT, not to mention MAD methods.

To gain a more rigorous and holistic view of MAD's performance relative to CoT, we aggregated results from 36 experimental configurations (four models, nine datasets). For each configuration, we conducted an ANOVA test with a significance level of $0.05$ to assess whether MAD frameworks statistically outperformed, tied, or underperformed compared to CoT. Based on the results, each comparison was categorized as a Win, Tie, or Lose. As shown in Figure 2, SoM, EoT, ChatEval, and AgentVerse only outperformed CoT in approximately 15% cases, while MP did not demonstrate significant improvement over CoT. Although ChatEval achieved the lowest loss rate, its win rate is still not greater than 15%. When examining performance by task type, MAD frameworks performed worse on programming tasks (only SoM/AgentVerse had positive win rates) but better on mathematical reasoning, surpassing their overall performance levels.

**Do we replicate previous results?** The empirical results presented above demonstrate that the considered MAD frameworks typically underperform the much simpler single-agent baseline CoT, a somewhat surprising finding that has not been reported before. However, in general, these MAD frameworks are able to outperform single-agent (SA) that are instructed to directly generate their answers. Specifically, we observe that MAD frameworks outperform SA in most conditions (34 out of 45 conditions utilizing GPT-4o-mini), which perfectly aligns with previous findings in the field.

We acknowledge that ChatEval and SoM were not compared to CoT though, EoT, AgentVerse, and MP were compared to CoT in their papers. However, MP was evaluated against CoT solely on the CIAR dataset [3], which was initially proposed and has not been fully disclosed. EoT was shown to outperform CoT [2]. Our evaluation replicates that EoT can surpass CoT on the GSM8K and CommensenseQA benchmarks, but this advantage was observed only when using Claude-3.5-haiku among the four models. Furthermore, Table 1 shows we also replicate AgentVerse's superior performance on the HumanEval benchmark, significantly outperforming other methods including CoT. However, note that on programming tasks, AgentVerse incorporates an additional execution-

---

[3] We follow [11] which assumes the need for a single correct answer to be determined by majority voting. As such, for SC, we exclude programming tasks that allow multiple valid programs.

evaluation stage so that agents can utilize the execution results from the generated programs, which is usually absent in other MAD frameworks and arguably beyond the scope of MAD designs.

**How do hyperparameters influence MAD performance?** As mentioned earlier, we by default followed the conventional choice of hyperparameters (the number of agents and the number of debate rounds). That said, one might be interested in how varying the choices influences MAD performance as well as our key findings.

Thus, we conducted a systematic ablation study, utilizing GSM8K, MMLU, and HumanEval as representative benchmarks for each top-level LLM capability. MAD frameworks with fixed numbers of agents, such as ChatEval and MP, were excluded from experiments varying the number of agents. Similarly, EoT, MP, and AgentVerse were excluded from experiments involving debate rounds due to their early stopping mechanism, which does not allow precisely adjusting the number of debate rounds.

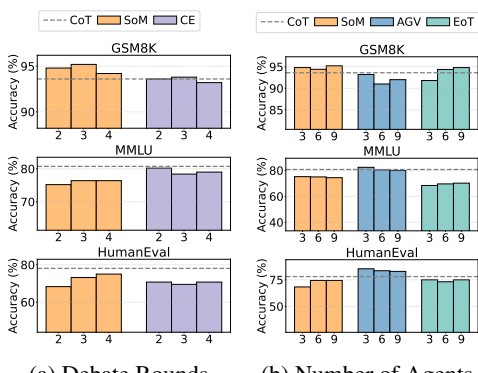

(a) Debate Rounds    (b) Number of Agents

Figure 3: We explore the impact of hyperparameters on the performance of MAD frameworks by increasing the debate rounds or number of agents, and compare the results to CoT.

Our empirical results, summarized in Figures 3a and 3b, indicate that in most scenarios, increasing the number of agents or debate rounds does not significantly change the outcomes. For instance, the SoM framework continues to underperform CoT on the MMLU and HumanEval benchmarks, even as debate rounds increase from 2 to 4 or the number of agents increases from 3 to 9 on HumanEval. Conversely, SoM always surpasses CoT on GSM8K when varying debate rounds. The sole notable exception is observed with EoT on the GSM8K benchmark, where increasing the number of agents from 3 to 9 leads to a continuous improvement in performance, ultimately surpassing CoT. Nonetheless, aside from this exception, increasing the number of agents or debate rounds often results in either stagnation or even a decline in performance. These results show that superior performance over CoT across various benchmarks cannot be realized by merely adjusting hyperparameters. Consequently, our study rules out suboptimal hyperparameter configurations as a universal explanation for the inferior performance of MAD when compared to CoT.

**Can MAD efficiently utilize more inference budget?** Beyond performance, we also assess the efficiency of MAD frameworks in utilizing inference-time computational resources. We measured the number of tokens consumed in our experiments. Unlike the number of LLM calls, which can be pre-defined, most LLMs do not allow precise control over token consumption. Moreover, for proprietary LLMs, the number of tokens consumed is typically positively correlated with cost. We present how the performance of MAD scales with the token consumption in Figure 5 in Appendix D. Note that we exclude MP in the figure as it must involve 2 agents and is not allowed to pre-define the number of debating rounds in advance due to its early-stopping mechanism. Moreover, for comparison, we additionally include the results of SC by increasing the number of samples it draws.

We observe that SC effectively utilizes the increased inference budget. However, MAD frameworks either: (i) show no positive trend in achieving stable performance improvements with more inference budget, e.g., SoM does not stably achieve better performance on MMLU as consuming more tokens, or (ii) continue to underperform SC while consuming a comparable number of tokens although positively scaling up, e.g., EoT performs better as more tokens consumed on MMLU and GSM8K, but still obviously underperforms SC or even other MAD frameworks. These observations indicate that, in comparison to SC, MAD is generally a less efficient method for leveraging token consumption.

**Why do MAD methods underperform single-agent baselines?** We analyze the performance of MAD on individual questions to gain deeper insights. Specifically, we compared each evaluated MAD method against SA by examining two key metrics: the number of incorrect answers corrected by the method and the number of correct answers erroneously altered by the method. Ideally, MAD should be able to correct a substantial number of errors while introducing minimal new errors. These results are visualized in Figure 6 in Appendix D.

Table 2: Performance results of Heter-MAD. CoT-Average represents the average performance achieved by these two models with CoT reasoning. We use  light green  to denote the highest performance achieved by a single MAD framework, and  green  the highest one overall. We record positive performance gain in red by comparing Heter-MAD to the average performance of MAD using two models, as well as CoT-Average.

| Dataset | MMLU | MMLU-Pro | CommensenseQA | ARC-Challenge | AGIEval | GSM8K | MATH | HumanEval | MBPP | Average |
|---|---|---|---|---|---|---|---|---|---|---|
| CoT-Average | 81.7±1.3 | 58.3±1.3 | 82.6±1.5 | 93.4±0.6 | 62.4±2.0 | 92.8±1.2 | 55.0±1.5 | 70.3±1.8 | 55.8±2.7 | 72.5±4.9 |
| SoM-GPT | 74.7 ± 0.5 | 62.8 ± 1.0 | 80.7 ± 0.9 | 90.8 ± 0.4 | 64.3 ± 0.3 | 94.9 ± 0.3 | 75.4 ± 0.7 | 68.1 ± 1.3 | 56.9 ± 1.1 | 74.3 ± 0.3 |
| SoM-Llama | 84.6 ± 0.4 | 57.1 ± 1.2 | 81.9 ± 0.3 | 92.9 ± 0.5 | 62.2 ± 1.1 | 88.3 ± 0.7 | 57.3 ± 0.3 | 63.4 ± 2.3 | 41.4 ± 0.5 | 69.9 ± 0.3 |
| SoM-Heter | 83.5 ± 0.1 | 65.0 ± 0.6 | 83.3 ± 0.1 | 92.1 ± 0.5 | 70.1 ± 0.4 | 94.6 ± 0.2 | 71.1 ± 0.9 | 75.8 ± 3.3 | 54.7 ± 1.9 | 76.7 ± 0.4 |
| *- vs SoM-Average* | 4.8% | 8.4% | 2.5% | 0.3% | 10.8% | 3.3% | 7.2% | 15.3% | 11.3% | +6.4% |
| *- vs CoT-Average* | 2.2% | 11.4% | 0.8% | -1.4% | 12.3% | 1.9% | 29.3% | 7.8% | -2.0% | +5.8% |
| EoT-GPT | 67.9 ± 0.4 | 61.2 ± 0.6 | 80.1 ± 0.5 | 86.4 ± 0.3 | 65.1 ± 0.7 | 94.4 ± 0.6 | 75.9 ± 1.2 | 73.8 ± 2.2 | 56.2 ± 0.5 | 73.4 ± 0.3 |
| EoT-Llama | 83.2 ± 0.3 | 49.7 ± 0.6 | 81.9 ± 0.7 | 93.0 ± 0.1 | 63.1 ± 0.5 | 77.6 ± 0.7 | 55.3 ± 0.3 | 55.5 ± 0.9 | 38.9 ± 1.7 | 66.5 ± 0.3 |
| EoT-Heter | 79.7 ± 4.4 | 63.6 ± 3.1 | 83.9 ± 0.5 | 92.7 ± 0.5 | 69.8 ± 0.5 | 93.2 ± 0.3 | 73.1 ± 0.6 | 70.9 ± 1.1 | 54.3 ± 0.4 | 75.7 ± 0.6 |
| *- vs EoT-Average* | 5.5% | 14.7% | 3.6% | 3.3% | 8.9% | 8.4% | 11.4% | 9.7% | 14.2% | +8.2% |
| *- vs CoT-Average* | -2.5% | 9.0% | 1.5% | -0.8% | 11.8% | 0.4% | 32.9% | 0.8% | -2.8% | +4.4% |
| ChatEval-GPT | 79.1 ± 0.9 | 62.2 ± 0.5 | 81.1 ± 0.8 | 93.2 ± 0.3 | 68.9 ± 0.9 | 93.6 ± 0.0 | 69.4 ± 1.6 | 71.8 ± 0.8 | 53.7 ± 0.6 | 74.8 ± 0.3 |
| ChatEval-Llama | 80.4 ± 1.2 | 56.1 ± 1.0 | 72.8 ± 1.3 | 89.9 ± 0.2 | 68.6 ± 0.4 | 92.5 ± 0.2 | 58.7 ± 1.5 | 62.8 ± 0.9 | 44.5 ± 2.2 | 69.6 ± 0.4 |
| ChatEval-Heter | 82.6 ± 0.5 | 64.9 ± 0.4 | 78.8 ± 1.4 | 92.3 ± 0.5 | 70.5 ± 0.7 | 94.6 ± 0.2 | 71.4 ± 0.7 | 70.9 ± 0.8 | 49.8 ± 2.2 | 75.1 ± 0.3 |
| *- vs CE-Average* | 3.6% | 9.7% | 2.4% | 0.8% | 2.5% | 1.7% | 11.5% | 5.3% | 1.4% | +4.0% |
| *- vs CoT-Average* | 1.1% | 11.3% | -4.6% | -1.2% | 13.0% | 1.9% | 29.8% | 5.3% | -10.8% | +3.6% |
| AgentVerse-GPT | 80.4 ± 0.0 | 62.1 ± 0.5 | 80.7 ± 0.4 | 92.5 ± 0.1 | 63.9 ± 1.2 | 92.7 ± 0.5 | 64.5 ± 1.4 | 85.4 ± 0.0 | 58.9 ± 0.2 | 75.7 ± 0.2 |
| AgentVerse-Llama | 84.8 ± 1.0 | 61.8 ± 0.9 | 76.5 ± 1.2 | 92.8 ± 0.3 | 66.7 ± 0.8 | 85.5 ± 0.7 | 45.3 ± 0.9 | 60.0 ± 0.8 | 41.9 ± 1.0 | 68.4 ± 0.3 |
| AgentVerse-Heter | 84.3 ± 1.0 | 63.0 ± 0.4 | 79.3 ± 0.8 | 92.6 ± 0.6 | 66.7 ± 1.0 | 90.7 ± 0.8 | 58.1 ± 0.2 | 78.5 ± 0.2 | 53.0 ± 0.2 | 74.0 ± 0.2 |
| *- vs AGV-Average* | 2.1% | 1.7% | 0.9% | -0.1% | 2.1% | 1.8% | 5.8% | 8.0% | 5.2% | +2.7% |
| *- vs CoT-Average* | 3.1% | 8.0% | -4.0% | -0.9% | 6.9% | -2.3% | 5.6% | 11.6% | -5.1% | +2.1% |

We find that MP, ChatEval, and AgentVerse, while capable of correcting many wrong answers, also frequently introduce a high number of misstatements by mistakenly altering the initially correct answers. This overly aggressive behavior prevents these methods from delivering stable and consistent improvements. On the other hand, methods such as SoM and EoT are more conservative, effectively limiting the frequency of errors while also reducing their ability to correct mistakes. These observations not only explain why the MAD methods typically fail to outperform CoT and SC, but also highlight a key trade-off in the MAD design: overly aggressive methods may introduce instability, while conservative methods may struggle to capitalize on opportunities for correction.

# 4  Improving the Status Quo: Model Heterogeneity as an Antidote

The previous section presents substantial evidence suggesting that we should stop overvaluing current MAD methods and rethink the prevailing evaluation protocols. While these negative results may seem discouraging, we argue that to fix this, MAD research must begin to embrace model heterogeneity as a core design principle. In this section, we present empirical evidence supporting this argument. Importantly, our goal is not to propose a fully optimized MAD framework built around model heterogeneity—such a contribution lies beyond the scope of this position paper. Instead, we demonstrate the potential of this direction through a simple modification to the MAD methods evaluated in the previous section.

## 4.1  Heterogeneous MAD

Intuitively, models trained on different data and paradigms may exhibit distinct strengths and weaknesses. Building on this idea, we posit that MAD designs leveraging model diversity can effectively compensate for individual model limitations while amplifying their strengths, ultimately leading to overall performance improvements.

To validate this hypothesis, we introduce **Heter-MAD**, a simple and general method that can be integrated into *any* existing MAD framework. Heter-MAD differs from existing MAD methods with only one key difference—every time that an agent generates an output, the agent queries a foundation model $i$ (where $i \in \{1, ..., n\}$) with probability $p_i$ (such that $\sum_{i=1}^{n} p_i = 1$) from a pool of candidate models. Therefore, Heter-MAD effectively reuses the prompts and architecture of any MAD method without requiring deliberate adjustments to incorporate different foundation models. This makes it well-suited for evaluating whether model heterogeneity can enhance MAD.

## 4.2  Experimental Results

We validate the effectiveness of Heter-MAD by considering GPT-4o-mini [28] and Llama3.1-70b [29] as candidate foundation models, with the probability of selecting each model simply setting to 0.5.

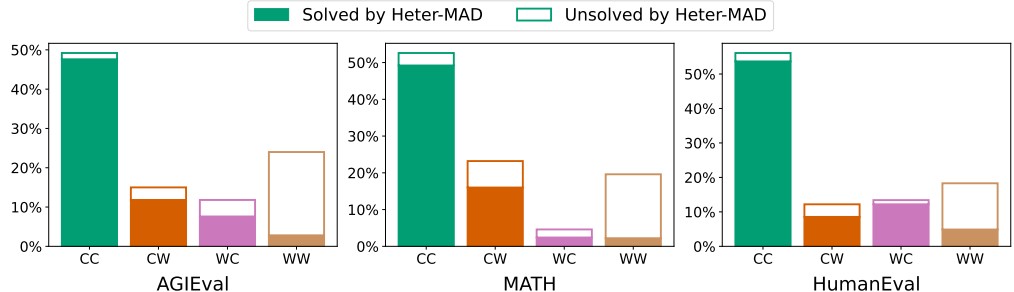

Figure 4: Heter-MAD performance analysis. Benchmark questions are categorized as: CC (both GPT-4o-mini and Llama3.1-70b solve), CW (only GPT-4o-mini solves), WC (only Llama3.1-70b solves), and WW (neither solves). Filled bars indicate questions solved by Heter-MAD; hollow bars indicate unsolved. For full results, see Figure 7 in Appendix D.

We instantiate Heter-MAD with SoM, EoT, ChatEval, and AgentVerse, while we exclude MP due to two reasons: (i) the agent roles in MP are unbalanced[3], and (ii) its performance is generally weak, achieving $0\%$ win rate compared to CoT.

**Heter-MAD consistently improves MAD** We present the performance results in Table 2. Notably, we find that Heter-MAD consistently improves the performance of all considered MAD frameworks. Specifically, by incorporating model heterogeneity, Heter-SoM improves SoM-average (the average performance achieved by SoM when utilizing the two candidate models separately) by 6.4%, and Heter-EoT improves EoT-average by 8.2%. Moreover, Heter-SoM, with SoM being the most simple and foundational MAD, achieves the highest performance, surpassing all the other, more recently developed MAD methods considered. Last but not least, by incorporating model heterogeneity, all the considered MAD methods outperform CoT-Average (the average performance achieved by CoT when utilizing the two candidate models separately) by up to 5.8%.

On the other hand, we observe that the performance gains brought by model heterogeneity on ChatEval and AgentVerse are also less significant compared to SoM or EoT, despite that ChatEval and AgentVerse represent more recent advancements. We hypothesize that the significant performance gain of Heter-SoM and Heter-EoT stems from the rather simple design of SoM and EoT, and thus they are more compatible with heterogeneous models. Conversely, more complex frameworks, like ChatEval and AgentVerse, struggle with compatibility when incorporating model heterogeneity and lack the ability to effectively aggregate the strengths of diverse models. This phenomenon highlights substantial opportunities for optimizing the compatibility of MAD frameworks with heterogeneous model ensembles. By enhancing the ability of MAD systems to integrate and leverage the strengths of varied models, future research can achieve more consistent and robust performance improvements, fully harnessing the potential of model heterogeneity in multi-agent collaborations.

**How Heter-MAD improves MAD?** To elucidate how model heterogeneity contributes to performance improvements, we conducted a detailed analysis of Heter-MAD's outcomes. As depicted in Figure 4, we categorized questions with their solvability by SoM-GPT and SoM-Llama into four groups: **CC** (both models correctly solve), **WW** (both models incorrectly solve), **CW** (SoM-GPT is correct while SoM-Llama is wrong), and **WC** (SoM-GPT is wrong while SoM-Llama is correct). Our observations reveal that the **CC** category constitutes the largest proportion of questions, and SoM-Heter consistently maintains high accuracy.

The primary booster of Heter-MAD's performance gains is its ability to handle **CW** and **WC** questions, which together account for a significant portion of the dataset. In these categories, Heter-MAD successfully leverages the strengths of each model, correcting errors that a single-agent baseline might miss. Conversely, for **WW** questions—where neither model can provide correct answers—Heter-MAD naturally exhibits low accuracy. However, the substantial improvements in the **CC**, **CW**, and **WC** categories sufficiently elevate the overall performance of Heter-MAD. This confirms that incorporating model heterogeneity enables MAD to leverage the diverse strengths of different models. By allowing agents to generate outputs using various models, this simple adjustment proves highly effective, significantly enhancing overall performance and paving the way for future research.

# 5 Key Questions for Future MAD Research

MAD research remains in its early stages, with many fundamental questions about the mechanisms driving effective multi-agent collaboration unexplored:

**How to fully leverage model heterogeneity in MAD?** Our empirical evaluation of Heter-MAD demonstrates that incorporating model heterogeneity within MAD frameworks is feasible and promising. By querying different foundation models, Heter-MAD has achieved notable improvements across most benchmarks. Additionally, we observed that SoM achieves the best performance when accessing a more cost-effective model, Llama3.1-70b, indicating that model heterogeneity can enhance performance while reducing computational costs. However, current MAD designs are not optimized for aggregating heterogeneous models effectively, as evidenced by the variable performance gains across different frameworks, as well as that SoM was the most foundational MAD framework without complex mechanisms. This highlights the need for developing more suitable MAD methods that can seamlessly integrate diverse models, thereby maximizing the benefits of model heterogeneity.

**How to enhance MAD frameworks with single-agent inference approaches?** While MAD primarily focuses on aggregating multiple agents for collaborative inference, empowering individual agents remains crucial. Correspondingly, we evaluate SoM and ChatEval, which do not explicitly incorporate CoT-style responses[4], in combination with CoT in Appendix D.1. We have two key findings: (i) CoT consistently improves MAD and Heter-MAD and (ii) Heter-MAD and MAD-CoT improve MAD in distinct directions. These results suggest that it is valuable to explore integrating more powerful single-agent inference approaches [30, 31], or advanced models with inherently strong reasoning capabilities [32, 33, 34], to further optimize collaborative inference outcomes.

**How to implement fine-grained interaction mechanisms?** Current MAD methods lack fine-grained interaction capabilities, as agents engage in debates based solely on their complete responses to a given query. This approach leads agents to emphasize the final answer, neglecting underlying reasoning steps. When responses diverge, rebuttals focus on outcome differences rather than analyzing the logic behind discrepancies. Future frameworks should prioritize agents that scrutinize reasoning processes, enabling debates targeting logical gaps to improve overall reasoning quality. A case study illustrating how agents debate when their answers differ is provided in Appendix B.

**What kind of application scenarios better reflect the utility of MAD?** Most existing benchmarks predominantly include test cases that require only a single knowledge point for resolution, suggesting that more advanced single-agent methods could suffice and that MAD frameworks may be unnecessary in these contexts. To illustrate this limitation, we provide a case study in Appendix B, demonstrating how simplistic benchmarks do not reflect the true potential of MAD in facilitating intricate reasoning processes. Consequently, it is essential to find scenarios that can better reflect the utility of MAD, e.g., scenarios naturally requiring diverse knowledge or capability from multiple agents.

# 6 Conclusions

This work critically examines the current landscape of MAD research, and highlight concerns regarding the reproducibility, efficiency, and generalizability of MAD research, emphasizing the need for more rigorous evaluation. Through systematic evaluation, we demonstrate that existing MAD frameworks fail to reliably outperform simple single-agent baselines like CoT and SC, despite consuming additional computational resources. This highlights an urgent need to reevaluate MAD research practices, particularly narrow benchmarks and inconsistent baselines that risk overstating collaborative benefits. We advocate for abandoning overvalued current MAD designs and fundamentally rethinking evaluation to ensure genuine progress.

Moreover, we propose several calls-to-action and potential directions to improve MAD, from robust evaluations to promoting knowledge and reasoning diversity via model heterogeneity—a particularly promising avenue. We intentionally avoid specific technical solutions, believing each direction warrants future research and aiming to spark broad community exploration. After all, if the age-old saying, 'two heads are better than one,' holds true, then collaboration—whether among human researchers or LLM agents—has the potential to make transformative advancements.

---

[4]EoT, MP, and AgentVerse have incorporated CoT-style answers in their designs.

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

Table 3: Evaluations details of previous MAD research. Overlapped benchmarks are marked out.

| | Benchmarks |
|---|---|
| SoM[1] | Arithmetic, **GSM8K**, Chess, Biographies, **MMLU**, Chess Move Validity |
| Multi-Persona[3] | CommonMT, CIAR |
| EoT [2] | **GSM8K**, MultiArith, SingleEQ, AddSub, AQuA, SVAMP |
| ChatEval [5] | LLM-evaluation, Topical-Chat |
| AgentVerse [4] | FED, Commengen-Challenge, MGSM, Logic Grid Puzzles, **HumanEval** |
| COMM [13] | College Physics, Moral Scenarios (two subsets from **MMLU**) |
| FORD [17] | $\alpha$NLI, **CSQA**, COPA, e-CARE, Social IQa, PIQA, StrategyQA |

# A  Background

## A.1  Related works

MAD methods have garnered significant attention in recent years due to their potential to enhance the reasoning and decision-making capabilities of LLMs. At their core, MAD methods share several common principles. In the initial design proposed by **SoM** [1], MAD typically involves multiple agents following three steps to generate the final response: (1) Response Generation, where each agent produces an initial solution based on its unique perspective; (2) Debate, where agents debate to identify logical inconsistencies or knowledge gaps; and (3) Consensus Building, where the consensus is determined by majority voting or a judge agent.

A series of following works explored enhancing the reasoning capabilities of MAD by assigning different roles to agents, enabling agents to debate from various perspectives. [12] explores the behavioral logic of MAD from the perspective of social psychology. The authors found that certain combinations of individual traits can enhance the overall performance of the MAD system. **Multi-Persona (MP)** [3] incorporates an affirmative agent (angel) and a negative agent (devil) presenting their answer to a judge agent, which ultimately determines the final solution. **Exchange-of-Thoughts (EoT)** [2] assigns three diverse roles to agents: detail-oriented nature, diligence, and problem-solving abilities. Additionally, it implements a confidence evaluation mechanism designed to reduce the adverse effects of erroneous reasoning processes. **COMM** [13] encourage diverse thinking in the debate by assigning different reasoning paths to agents with different roles.

A part of the research focused on improving communication topology. **IoA** [14] organizes agents in a network structure, splitting agents into blocks for better collaboration. In [15], agents communicate through a sparse topological structure. In **AgentVerse** [4], the verifier can dynamically determine the subsequent execution of MAD processing, allowing dynamic adjustment of communication topology. MacNet [16] investigated the scaling effects of MAD systems with more agents using varied communication structures, and found that MAD systems can achieve consistent performance improvements with more agents involved.

Another line of work enhanced the way to exchange and integrate information between agents. **FORD** [17] mitigates the inconsistency as the debate processes, and introduces a judge agent to summarize the debate results. **ReConcile** [18] adopts confidence-weighted voting to help consensus seeking. [35] introduced a novel approach where agents interact using token embeddings instead of natural language. **ChatEval** [5] explores communication strategies among agents through three frameworks, focusing on the impact of asynchronous responses and round-by-round summarization on agent performance.

With the emergence of an increasing number of MAD frameworks, some recent studies have reviewed various MAD methods from different aspects. [7] found that MAD methods do not reliably outperform other ensembling reasoning strategies. However, they specifically focus on medical prompting methods and medical benchmarks, which limits the generalizability. [8] analyzed the performance of MAD systems from the perspective of persuasiveness and found that more persuasive models could enhance the overall MAD performance. [6] compared single-agent methods with MAD methods and found that providing sufficiently detailed problem descriptions can enhance single-agent inference to a level comparable to MAD methods. However, the single-agent inference approach used in the comparison was specifically calibrated for these detailed descriptions, rather than being a widely adopted single-agent method. Additionally, the evaluation was limited to only three datasets.

In summary, while there are positive results celebrating MAD, there are also recent efforts questioning whether MAD is a reliable general approach for enhancing LLM performance. However, limitations in their evaluation leave the answer unresolved and inconclusive. This underscores the urgent need for a more thorough and comprehensive evaluation, and necessitates rethinking common evaluation practices in MAD research, particularly the reliance on narrow benchmarks and inconsistent baselines.

## A.2  Flaws in previous evaluation

We summarize evaluation details of previous MAD research in Table 3 as complementary background information to support our previous claim on improper evaluation of MAD frameworks. Specifically, we observe there were minimal overlap between benchmarks in previous research. To provide context for our claim, we draw upon benchmarks considered in prior research, but not featured in the main text, as follows.

**Arithmetic**  evaluates the ability of models to correctly evaluate an arithmetic expression (containing addition, multiplication, and subtraction) consisting of six different two-digit numbers. This dataset contains randomly generated mathematical expressions.

**Chess**  evaluates the strategic reasoning ability of models. Models are asked to predict the best next move in a game of chess, given the first 14 moves of a chess game between two chess grand-masters.

**Biographies**  tests the factuality of language models. Models are asked to accurately generate historical biographies of people. This benchmark contains 524 well-known people with ground truth bullet-point biographies.

**Chess Move Validity**  checks whether models can avoid hallucination in following given rules. Specifically, this benchm ark measure the validity of possible moves in a game of Chess given by BIG-Bench Chess-State Tracking Benchmark [36]. The evaluated language agent is given a set of next moves and required to find a valid move as the answer.

**CommonMT (Commonsense Machine Translation)** [37] contains Chinese to English translation examples. Accurate translation requires an understanding of common sense knowledge to distinguish ambiguous expressions.

**CIAR (Counter-Intuitive Arithmetic Reasoning)**  is proposed to evaluate the reasoning abilities of LLMs at deep levels. This benchmark is designed to be resistant to intuitive reasoning, and requires multi-step reasoning.

**MultiArith** [38] is composed of a diverse set of multi-step arithmetic word problems that require understanding complex relationships between quantities and performing multiple operations to arrive at the correct numerical answer.

**SingleEQ** [39] is a collection of algebraic word problems designed for evaluating systems that can translate these problems into solvable mathematical equations.

**AddSub** [40] is a simple dataset designed for evaluating models on arithmetic word problems that involve addition and subtraction operations.

**AQuA** [41] contains 100,000 algebraic word problems presented in a multiple-choice format, uniquely featuring natural language rationales that detail the step-by-step reasoning to arrive at the correct solution.

**SVAMP** [42] is created by applying controlled variations to existing elementary-level arithmetic word problems (typically grade four and below) to test the robustness and true understanding of models.

**LLM-evaluation** [43] investigates the potential of utilizing LLMs as a substitute for human evaluators in assessing the quality of machine-generated text. LLMs are provided with the same instructions, text samples, and questions that human evaluators would receive.

**Topical-Chat** [44] contains human-human conversations where participants discuss topics based on provided knowledge snippets covering eight broad categories, without pre-defined roles.

**FED** [45] offers an automated, unsupervised method for assessing open-domain dialogue systems. It uniquely leverages a pre-trained DialoGPT model, allowing it to evaluate aspects like fluency and coherence.

732 **Commengen-Challenge** [46] requires the model to compose a coherent sentence describing an
733 everyday scenario using a given set of concepts. This task is particularly challenging as it requires
734 models to perform relational reasoning with background commonsense knowledge and exhibit
735 compositional generalization for novel concept combinations.

736 **MGSM** [47] consists of 250 grade-school math problems from the GSM8K dataset, which were
737 manually translated into ten diverse languages, including those with fewer resources like Bengali and
738 Swahili.

739 **Logic Grid Puzzles** [48] targets the logical reasoning abilities of models by requiring them to solve
740 puzzles where they must deduce relationships between different categories of items based on a series
741 of clues.

742 # B Case study

743 **Limited benchmark** Most test cases in existing benchmarks only require a single knowledge point
744 to be solved, which naturally makes MAD unnecessary. Here we present a case study to support our
745 argument. We pick some test cases from MMLU, MMLUPro, and CommensenseQA as follows.

---

**MMLU, college_physics**
Which of the following gives the total spin quantum number of the electrons in the ground
state of neutral nitrogen (Z = 7)?
A: "1/2" B: "1" C: "3/2" D: "5/2"
**MMLUPro**
A television sells for $180. It costs the retailer $136.50. Find the percent markup based on
the selling price.
A: "27.5%" B: "21%" C: "22.5%" D: "30%" E: "24.17%" F: "20%" G: "26.5%" H: "25%" I:
"18%" J: "32.5%"
**CommensenseQA**
The child's wild imagination made him able to see story that he read, he was able to do what
with the story?
A: "picture it" B: "reading" C: "visualize" D: "open book" E: "go to movies"

---

747 **Agents debate based on whether their answers are the same** In practice, we found that agents'
748 behavior in debate relies on the fact that their answers are the same (or not).

---

**Question:** What is the feeling of one having fun?
**Round 1**
Agent 1: B
Agent 2: E: laughter
Agent 3: The answer is B
**Round 2**
Agent 1: Considering the solutions provided by other agents, the feeling of having fun is often
associated with joy, pleasure, and expressions like laughter. While "pleasure" (B) reflects the
overall enjoyment, "laughter" (E) specifically captures a direct expression of fun. Given that
laughter is a common reaction to fun situations, I would align with that sentiment. Thus, the
updated answer is (E).
Agent 2: Considering the additional opinions from other agents, they suggested that the feeling
of one having fun corresponds to "pleasure" (B). While laughter (E) is often associated with
fun, the broader concept of having fun can also encompass feelings of pleasure. Therefore, I
will incorporate these insights into my final answer. The updated answer is (B).
Agent 3: Considering the feeling of having fun, both pleasure and laughter are associated
with enjoyable experiences. While pleasure (B) is a broader term that encompasses a range of
enjoyable feelings, laughter (E) is a direct expression that typically accompanies fun. Given
that one agent has chosen laughter as the answer and it specifically denotes a physical reaction
to joy, I will align with that insight. Therefore, my updated answer is (E).

---

Note that in these case studies, determining whether a problem requires only a single knowledge point to solve, or whether the agent overly relies on the similarity or difference of answers, is challenging to achieve objectively. Therefore, we use case studies to provide factual support for the future directions discussed in the paper, rather than analyzing this through statistical methods. We understand that this approach might make our argument less convincing, but it still aligns with our empirical observations. We also hope that future work will develop better methods to address and present these issues.

# C   Experiment details

## C.1   Benchmark configurations

Table 4: Benchmark configurations

| Benchmark | Category | Metric |
|---|---|---|
| MMLU | General Knowledge | accuracy, 0-shot |
| MMLUPro | General Knowledge | accuracy, 0-shot |
| CommensenseQA | General Knowledge | accuracy, 0-shot |
| ARC-Challenge | General Knowledge | accuracy, 0-shot |
| AGIEval | General Knowledge | accuracy, 0-shot |
| GSM8k | Mathematical Reasoning | accuracy, 0-shot |
| MATH | Mathematical Reasoning | accuracy, 0-shot |
| HumanEval | Programming | Pass@1, 0-shot |
| MBPP | Programming | Pass@1, 0-shot |

**MMLU** [19, 49] is a benchmark dataset designed to evaluate general knowledge across 57 subjects, including STEM, humanities, and social sciences. It tests a model's ability on challenging multiple-choice questions that require both specialized and common knowledge. The testing set contains 14,042 samples.

**MMLUPro** [20] is a more robust and challenging massive multi-task understanding dataset tailored to more rigorously benchmark LLMs' capabilities. It contains 1,200 testing samples.

**CommonSenseQA** [22] is a multiple-choice question-answering dataset that tests different types of commonsense reasoning. The dataset contains 1,140 questions with one correct answer and four distractor answers.

**AGi-Eval** [21] is a dataset aimed at evaluating artificial general intelligence (AGI) capabilities, focusing on problem-solving, reasoning, and generalization across multiple domains. We specifically use several subsets: *aqua-rat*, *logiqa-en*, *lsat-ar*, *lsat-lr*, *lsat-rc*, *sat-math*, *sat-en*, and *sat-en-without-passage*, as they are in English.

**ARC-Challenge** [23] contains genuine grade-school level, multiple-choice science questions. The dataset is partitioned into a Challenge Set and an Easy Set and has 3,548 questions in total.

**GSM8k** [24] is a challenging math word benchmark designed to test a model's reasoning and problem-solving abilities in arithmetic and algebra. It is widely used for evaluating mathematical understanding and reasoning. We use the main set of GSM8k, which contains 1,319 testing cases.

**MATH** [25] is a comprehensive dataset featuring math problems across various topics, including geometry, algebra, calculus, and number theory. The dataset contains 5,000 testing questions.

**HumanEval** [26] is a dataset for evaluating code generation and programming skills. It contains prompts and corresponding solutions for coding tasks. The dataset consists of 164 programming problems.

**MBPP** [27] is a programming dataset. MBPP is harder to solve than HumanEval since it does not include a function signature for reference. It contains 500 samples for evaluation.

We use the following prompts to help LLMs format their responses.

> **Multi-choice benchmarks**
> Instruction: Answer this multiple choice question. Generate your final answer by the answer is (X).
> Q: {question}
> A: The answer is
> **Mathematical reasoning benchmarks**
> Instruction: Answer this question. Generate your final answer by the answer is $\boxed{ANSWER}$.
> Q: {question}
> A: The answer is
> **Programming benchmarks**
> Instruction: Write a python program to complete the following code. Do not output any example usage. Generate the final program by "The answer is: '''python
> Q: {question}
> A:

## C.2 MAD configurations

Table 5: High-level comparison of MAD frameworks.

|  | Role-Play | Answer Generation | #Agents | #Rounds | Post-procesisng | Heterogeneous |
|---|---|---|---|---|---|---|
| SoM | N/A | Majority Voting | Adjustable | Fixed | N/A | No |
| MP | Fixed | Judger | Fixed | Early-stopping | N/A | Yes |
| EoT | Fixed | Majority Voting | Adjustable | Early-stopping | Confidence | No |
| ChatEval | Fixed | Majority Voting | Adjustable | Early-stopping | N/A | Yes |
| AgentVerse | Dynamic | Judger | Adjustable | Early-stopping | N/A | Yes |

**Society of Minds** 3 agents debate for 2 rounds. All agents share the same prompt as follows.

> These are the solutions to the problem from other agents:
> One agent's solution: {}
> One agent's solution: {}
> One agent's solution: {}
> Use these opinions carefully as additional advice, can you provide an updated answer? Make sure to state your answer (capital multiple choice letter) at the end of the response.

**MP** The angel agent first generates a solution, and the devil agent debates against the solution and presents a new one. The judger agent can continue the debate for another round, or end the debate by picking the final solution from these two solutions. The debate can continue for 5 rounds if the judger has not picked the final answer.

> **Angel's prompt:**
> You will now think step by step and provide an answer at the end of your response.
> **Devil's prompt:**
> You disagree with my answer. Provide your answer and reasons.
> **Judger's prompt:**
> You, as the moderator, will evaluate both sides' answers and determine if there is a clear preference for an answer candidate. If so, please summarize your reasons for supporting affirmative/negative side and give the final answer that you think is correct, and the debate will conclude. If not, the debate will continue to the next round. Now please output your answer in json format, with the format as follows: {"Whether there is a preference": "Yes or No", "Supported Side": "Affirmative or Negative", "Reason": "", "debate_answer": "the capital letter corresponding to the answer"}. Please strictly output in JSON format, do not output irrelevant content.

**Exchange of Thoughts** In EoT, three agents with diverse persona prompts are organized to perform the debate. Each agent can access other agents' response and their confidences. EoT adopts answer

consistency as the confidence signal, i.e., the frequency of the most frequent answer. These agents can debate at most 5 rounds. When all agents reach a consensus, the debate terminates.

---

Here is a solution process from your friend:
{}'s solution: {} {}'s confidence in this solution is: {}
{}'s solution: {} {}'s confidence in this solution is: {}
{}'s solution: {} {}'s confidence in this solution is: {}
Based on your friend's solution, carefully re-examine your previous answer. If your friend's confidence level is below 0.5, it suggests a high probability that the solution might be incorrect. Remember, solutions with high confidence can also be wrong. Utilize your talent and critical thinking to provide a new step-by-step solution process.
Provide the new solution directly, refrain from commenting on your friend's approach, and conclude by stating the answer.

**Kitty's persona prompt:**
You are Kitty, a high school student admired for your attentiveness and detail-oriented nature. Your friends often rely on you to catch details they might have missed in their work. Your task is to carefully analyze the presented math problem, apply your attentive skills, and piece together a detailed solution. Afterward, you'll have the opportunity to review the solutions provided by your friends, offering insights and suggestions. Your careful revisions will help all of you to enhance your understanding and arrive at the most accurate solutions possible.

**Ben's persona prompt:** You are Ben, a high school student with a track record of excellent grades, particularly in mathematics. Your friends admire your diligence and often seek your guidance in their studies. Your role is to scrutinize the problem at hand with your usual attention to detail, drawing from your vast knowledge of math principles. After considering your friends' approaches, carefully construct your answer, ensuring to clarify each step of your process. Your clear and logical explanations are valuable, as they will serve as a benchmark for your friends to compare and refine their own solutions.

**Peter's persona prompt:**
You are Peter, a high school student recognized for your unique problem-solving abilities. Your peers often turn to you for assistance when they encounter challenging tasks, as they appreciate your knack for devising creative solutions. Today, your challenge is to dissect the given math problem, leveraging your unique problem-solving strategies. Once you've crafted your solution, share it with your friends, Ben and Kitty, so they can see a different perspective. Your innovative approach will not only provide an answer but also inspire Ben and Kitty to think outside the box and possibly revise their own solutions.

---

**ChatEval** In ChatEval, three agents General Public, Critic, and Scientist debate one by one. Different persona prompts enable these agents to think in diverse style, with specific focus on critical thinking or scientific domain background. The debate continues for 2 rounds by default.

---

**General Public's prompt:**
We would like to request your answer to this question.
There are a few other referee assigned the same task, it's your responsibility to discuss with them and think critically before you make your final judgement.
You are now General Public, one of the referees in this task. You are interested in the story and looking for updates on the investigation. Please think critically by yourself and note that it's your responsibility to answer the question.
Now it's your time to talk, please make your talk short and clear, General Public!

**Critic's prompt:**
We would like to request your answer to this question.
There are a few other referee assigned the same task, it's your responsibility to discuss with them and think critically before you make your final judgement.
You are now Critic, one of the referees in this task. Your job is to question others judgement to make sure their judgement is well-considered.
Now it's your time to talk, please make your talk short and clear, Critic!

**Scientist's prompt:**
We would like to request your answer to this question.

---

> There are a few other referee assigned the same task, it's your responsibility to discuss with them and think critically before you make your final judgement.
> You are Scientist, one of the referees in this task. You are a professional engaged in systematic study who possesses a strong background in the scientific method, critical thinking, and problem-solving abilities. Please help other people to answer the question.
> Now it's your time to talk, please make your talk short and clear, Scientist!

**AgentVerse** AgentVerse adopts a dynamic way to organize the debate. First, a Rold Assigner agent reads the question and determines what kinds of agents should be recruited to solve the question. After the role assignment, one solver agent and three critic agents with assigned roles will debate to figure out the answer. Finally, an evaluator will review the answer and determine whether another round is necessary for a better answer. In dealing with programming tasks, an extra executor will be incorporated to execute the written program and return the execution result to the evaluator for accurate feedback.

---

**Role Assigner's prompt:**
\# Role Description
You are the leader of a group, now you are facing a problem:
{question}
You can recruit {cnt_critic_agents} people to solve the logic problem. What people will you recruit?
Here are some suggestion: {advice}
Response Format Guidance
You should respond with a list of expert description. For example:
1. an electrical engineer specified in the filed of xxx.
2. an economist who is good at xxx.
3. a lawyer with a good knowledge of xxx.
...
Only respond with the description of each role. Do not include your reason.

**Solver's prompt:**
Using these information, can you provide the correct solution to the problem? Explain your reasoning and solve the problem step by step. Your final answer should be a single capital letter, which is the lable of choice, in the form
boxedanswer, at the end of your response.

**Critic's prompt:**
You are in a discussion group, aiming to collaborative solve the following logic problem:
{question}
You are {role_description}. Based on your knowledge, can you check the correctness of the solutions given above? You should give your correct solution to the problem step by step. When responding, you should follow the following rules:
1. Double-check the above solutions, give your critics, then generate the correct solution step by step.
2. If the final answer in your solution is the same as the final answer in the above provided solution, end your response with a special token "[Agree]".
3. You must highlight your final answer in the form \boxed{answer} at the end of your response. The answer must be a single letter.
Now give your response.

**Executor's prompt:**
You are an experienced program tester. Now your team is trying to solve the problem:
Complete the Python function:
{question}
The solution has been written to 'tmp/main.py'. Your are going to write the unit testing code for the solution. You should respond in the following format:
Thought: your thought
Reasoning: your reasoning on the testing cases
Criticism: constructive self-criticism
File Path: the path to write your testing code

Code: the testing code with explaination in docstring. make sure to write the input in the assertion to make it appear in the unit test report, and make sure the expected answer is correct

Command: the command to change directory and execute your testing code

**Evaluator's prompt:**

Problem:

{question}

Solution:

{solution}

You are a logic problem lover. Above is a logic problem and a solution. Check whether the solution and the deduction is correct. If the deduction is wrong, you should explain why it is wrong, but do not give your solution. When it is correct, output a correctness of 1 and why it is correct.

You should respond in the following format:

Correctness: (0 or 1, 0 is wrong, and 1 is correct)

Response: (explain in details why it is wrong or correct. do not provide your solution)

# D  Additional Experimental Results

Table 6: Main results on Llama3.1:8b. We use lightred / lightblue to denote results higher/lower than CoT.

| Dataset | MMLU | MMLU-Pro | CommensenseQA | ARC-Challenge | AGIEval | GSM8K | MATH | HumanEval | MBPP |
|---|---|---|---|---|---|---|---|---|---|
| SA | 43.13 ± 1.04 | 34.27 ± 0.50 | 66.00 ± 0.16 | 81.67 ± 0.68 | 33.60 ± 0.75 | 70.40 ± 0.43 | 36.47 ± 1.09 | 54.07 ± 0.29 | 50.45 ± 1.75 |
| CoT | 57.47 ± 1.18 | 41.20 ± 1.14 | 71.13 ± 0.84 | 86.40 ± 0.99 | 46.73 ± 1.09 | 80.13 ± 1.23 | 40.13 ± 0.66 | 37.60 ± 1.52 | 43.71 ± 2.88 |
| SC | 64.96 ± 1.08 | 47.49 ± 0.08 | 74.43 ± 0.30 | 86.60 ± 1.13 | 42.47 ± 1.58 | 79.53 ± 0.68 | 42.25 ± 2.15 | – | – |
| SoM | 53.40 ± 0.28 | 36.57 ± 1.27 | 70.93 ± 0.82 | 82.00 ± 0.65 | 37.13 ± 0.47 | 63.87 ± 0.93 | 40.20 ± 0.85 | 47.56 ± 2.28 | 45.91 ± 1.15 |
| MP | 53.33 ± 2.54 | 36.44 ± 2.74 | 46.07 ± 0.62 | 61.93 ± 1.39 | 44.50 ± 1.65 | 47.60 ± 2.73 | 10.30 ± 0.93 | 24.19 ± 3.67 | 23.09 ± 1.81 |
| EoT | 48.97 ± 0.58 | 36.04 ± 0.21 | 66.15 ± 0.61 | 81.60 ± 0.43 | 33.42 ± 0.84 | 61.87 ± 2.39 | 26.61 ± 1.53 | 22.36 ± 2.07 | 23.87 ± 0.97 |
| ChatEval | 61.81 ± 0.88 | 43.56 ± 1.22 | 68.66 ± 2.62 | 84.70 ± 0.51 | 57.87 ± 1.25 | 81.13 ± 0.81 | 39.77 ± 0.54 | 41.46 ± 0.00 | 40.73 ± 1.60 |
| AgentVerse | 13.27 ± 0.47 | 20.53 ± 1.23 | 16.33 ± 1.52 | 24.60 ± 0.98 | 50.33 ± 2.49 | 5.47 ± 1.31 | 13.30 ± 1.26 | 40.24 ± 2.59 | 32.56 ± 1.32 |

Table 7: Main results on Llama3.1:70B. We use lightred / lightblue to denote results higher/lower than CoT.

| Dataset | MMLU | MMLU-Pro | CommensenseQA | ARC-Challenge | AGIEval | GSM8K | MATH | HumanEval | MBPP |
|---|---|---|---|---|---|---|---|---|---|
| SA | 80.20 ± 2.05 | 46.27 ± 0.66 | 79.13 ± 1.05 | 91.67 ± 0.25 | 56.87 ± 1.93 | 69.47 ± 0.90 | 38.13 ± 1.09 | 63.41 ± 1.72 | 45.78 ± 2.95 |
| CoT | 82.73 ± 1.25 | 53.87 ± 0.90 | 82.40 ± 1.45 | 93.33 ± 0.50 | 58.42 ± 1.53 | 92.07 ± 0.90 | 37.13 ± 0.98 | 62.60 ± 1.04 | 49.42 ± 2.52 |
| SC | 83.73 ± 0.19 | 53.27 ± 1.06 | 81.16 ± 0.63 | 92.80 ± 0.59 | 61.07 ± 0.25 | 83.20 ± 0.33 | 49.73 ± 0.62 | – | – |
| SoM | 84.60 ± 0.43 | 57.13 ± 1.15 | 81.93 ± 0.34 | 92.93 ± 0.52 | 62.22 ± 1.08 | 88.27 ± 0.74 | 48.64 ± 1.23 | 63.41 ± 2.28 | 41.37 ± 0.49 |
| MP | 81.39 ± 1.09 | 51.93 ± 2.29 | 68.55 ± 1.02 | 88.38 ± 0.03 | 61.60 ± 2.55 | 69.27 ± 1.05 | 24.60 ± 1.56 | 52.64 ± 1.04 | 32.56 ± 0.97 |
| EoT | 83.20 ± 0.28 | 49.66 ± 0.60 | 81.87 ± 0.74 | 92.96 ± 0.14 | 63.06 ± 0.46 | 77.60 ± 0.65 | 43.63 ± 2.28 | 55.49 ± 0.86 | 38.91 ± 1.68 |
| ChatEval | 80.37 ± 1.15 | 56.13 ± 1.00 | 72.82 ± 1.33 | 89.94 ± 0.20 | 68.59 ± 0.42 | 92.53 ± 0.19 | 58.73 ± 1.52 | 62.80 ± 0.86 | 44.49 ± 2.23 |
| AgentVerse | 84.80 ± 1.02 | 61.80 ± 0.91 | 76.47 ± 1.24 | 92.80 ± 0.28 | 66.73 ± 0.84 | 85.47 ± 0.68 | 45.33 ± 0.94 | 59.96 ± 0.76 | 41.89 ± 1.02 |

Table 8: Main results on Claude-3.5-Haiku. We use lightred / lightblue to denote results higher/lower than CoT.

| Dataset | MMLU | MMLU-Pro | CommensenseQA | ARC-Challenge | AGIEval | GSM8K | MATH | HumanEval | MBPP |
|---|---|---|---|---|---|---|---|---|---|
| SA | 56.81 ± 0.15 | 38.38 ± 0.36 | 79.40 ± 0.28 | 87.17 ± 0.54 | 48.99 ± 1.31 | 83.13 ± 0.09 | 31.71 ± 2.48 | 66.26 ± 0.76 | 48.55 ± 0.62 |
| CoT | 62.00 ± 0.00 | 47.00 ± 1.57 | 79.67 ± 0.34 | 89.47 ± 0.38 | 52.00 ± 1.02 | 85.84 ± 0.72 | 30.62 ± 1.42 | 65.24 ± 2.59 | 56.16 ± 0.92 |
| SC | 63.08 ± 1.06 | 50.02 ± 1.47 | 81.27 ± 0.05 | 90.00 ± 0.28 | 53.59 ± 1.09 | 90.27 ± 0.46 | 35.09 ± 0.69 | – | – |
| SoM | 57.39 ± 1.00 | 39.91 ± 0.47 | 79.40 ± 0.59 | 88.20 ± 0.23 | 51.48 ± 0.42 | 86.87 ± 0.73 | 34.31 ± 0.89 | 65.33 ± 1.15 | 58.09 ± 1.35 |
| MP | 55.68 ± 0.50 | 42.33 ± 1.65 | 55.39 ± 1.37 | 79.72 ± 0.74 | 46.54 ± 2.53 | 51.15 ± 2.48 | 12.01 ± 0.26 | 60.08 ± 2.61 | 51.84 ± 0.89 |
| EoT | 57.30 ± 0.75 | 39.15 ± 0.59 | 79.70 ± 0.35 | 87.41 ± 0.31 | 50.44 ± 0.49 | 87.00 ± 0.49 | 33.08 ± 1.93 | 66.33 ± 0.35 | 58.13 ± 1.92 |
| ChatEval | 58.40 ± 0.17 | 43.87 ± 0.51 | 70.97 ± 1.61 | 83.68 ± 0.25 | 53.00 ± 1.27 | 85.75 ± 0.61 | 30.76 ± 0.30 | 52.44 ± 1.00 | 46.69 ± 1.46 |
| AgentVerse | 62.85 ± 0.75 | 47.72 ± 1.63 | 78.27 ± 0.68 | 89.66 ± 0.56 | 56.16 ± 0.82 | 59.38 ± 0.69 | 30.73 ± 1.69 | 45.68 ± 8.57 | 43.22 ± 2.56 |

## D.1  Enhancing current MAD frameworks with stronger single-agent inference approaches

As discussed in Section 5, enhancing current MAD frameworks with stronger single-agent inference methods represents a valuable future direction. We note that while EoT, AgentVerse, and MP incorporate CoT-like mechanisms, SoM and ChatEval do not explicitly prompt agents to respond in a CoT style. Therefore, we evaluate SoM and ChatEval combined with CoT, to investigate the impact of stronger single-agent inference approaches on the overall performance of MAD frameworks.

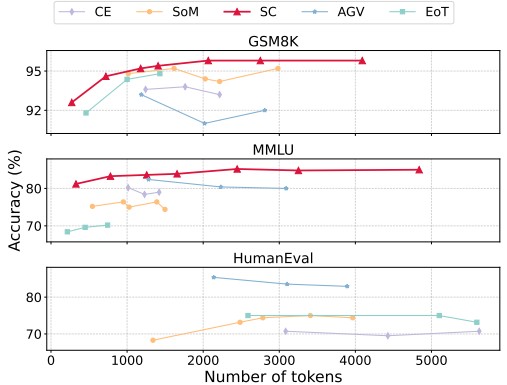

Figure 5: Comparing scaling efficiency of MAD methods. We present performance regarding number of tokens consumed.

Table 9: Empirical results on GPT-4o-mini combing CoT. We use lightred / lightblue to denote results higher/lower than CoT.

| Dataset | MMLU | MMLU-Pro | CommensenseQA | ARC-Challenge | AGIEval | GSM8K | MATH | HumanEval | MBPP |
|---|---|---|---|---|---|---|---|---|---|
| SA | 65.33 ± 0.93 | 58.07 ± 0.50 | 79.47 ± 0.25 | 88.27 ± 0.41 | 63.87 ± 1.05 | 91.13 ± 0.34 | 71.67 ± 1.31 | 66.67 ± 1.15 | 58.11 ± 0.66 |
| CoT | 80.73 ± 0.34 | 62.80 ± 0.99 | 82.87 ± 0.25 | 93.53 ± 0.41 | 66.40 ± 1.30 | 93.60 ± 0.82 | 72.87 ± 1.20 | 78.05 ± 1.49 | 62.26 ± 0.84 |
| SC | 82.13 ± 0.66 | 66.27 ± 1.39 | 83.80 ± 0.28 | 93.93 ± 0.25 | 67.07 ± 0.84 | 95.67 ± 0.19 | 73.96 ± 0.54 | – | – |
| SoM | 74.73 ± 0.52 | 62.80 ± 1.02 | 80.73 ± 0.93 | 90.80 ± 0.43 | 64.33 ± 0.34 | 94.93 ± 0.34 | 75.40 ± 0.71 | 68.09 ± 1.25 | 56.94 ± 1.12 |
| +CoT | 84.13 ± 0.50 | 65.26 ± 1.90 | 83.33 ± 0.31 | 93.67 ± 0.23 | 66.84 ± 0.74 | 94.67 ± 0.42 | 75.40 ± 0.53 | 78.86 ± 1.27 | 61.22 ± 0.59 |
| +CoT +Heter | 87.27 ± 0.47 | 68.20 ± 0.5 | 84.81 ± 0.61 | 93.75 ± 0.45 | 70.03 ± 0.58 | 96.00 ± 0.25 | 74.80 ± 1.05 | 78.66 ± 3.05 | 59.92 ± 1.16 |
| ChatEval | 79.13 ± 0.90 | 62.20 ± 0.49 | 81.07 ± 0.84 | 93.20 ± 0.28 | 68.87 ± 0.94 | 93.60 ± 0.00 | 69.36 ± 1.58 | 71.75 ± 0.76 | 53.70 ± 0.55 |
| +CoT | 82.40 ± 0.40 | 64.13 ± 0.64 | 84.67 ± 0.95 | 93.65 ± 0.31 | 65.87 ± 0.81 | 95.40 ± 0.40 | 70.60 ± 1.40 | 77.40 ± 3.05 | 60.96 ± 0.22 |

Particularly, we explicitly prompt each agent in MAD to respond in a CoT style. Our experimental results are shown in Table Table 9, from which we have several key findings

- **CoT consistently improves MAD and Heter-MAD.** MAD-CoT and Heter-MAD-CoT surpass CoT on all benchmarks except MBPP. Notably, on MMLU, SoM-CoT improves CoT by 3.4% and vanilla SoM by 9.4%, while Heter-SoM-CoT improves CoT by 6.54% and vanilla SoM by 12.54%. We can also observe obvious improvements for ChatEval, where ChatEval-CoT also surpasses CoT on 5 benchmarks.

- **Heter-MAD and MAD-CoT improve MAD in distinct directions.** Interestingly, we observe that Heter-SoM-CoT only achieves approximate performances in comparison to SoM-CoT on mathematical reasoning and programming tasks (e.g., Heter-SoM-CoT only outperforms SoM-CoT on GSM8k among four benchmarks). However, Heter-SoM-CoT consistently outperforms SoM-CoT on all five general knowledge tasks, showing a significantly different trend. This observation suggests that Heter-MAD and MAD-CoT improve MAD from distinct perspectives, and they can work compatibly.

Our findings provide initial insight that integrating multi-agent collaborative inference methods with enhanced single-agent inference approaches can be a promising future direction. Notably, while CoT is generally compatible, it remains unclear whether other advanced single-agent inference approaches, such as ReACT [30] or Reflexion [31], would achieve similar improvements, as these methods may alter the original behavior of single agents. We leave this for future investigation.

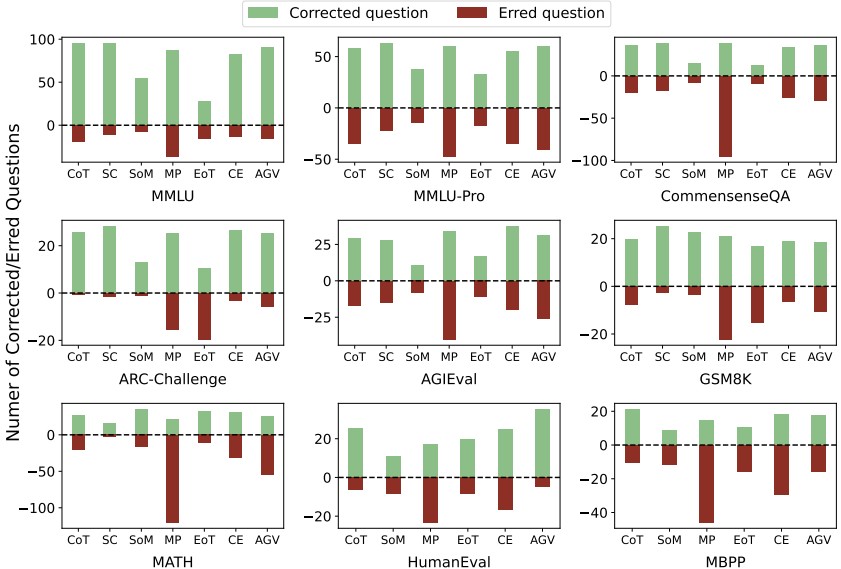

Figure 6: Comparing the behavior of inference strategies to direct prompting a single-agent. The green bar represents the number of corrected answers, and the Fred bar represents the number of answers erroneously reversed compared to standard single-agent prompting.


Figure 7: Heter-MAD performance analysis. We split questions in a benchmark into four parts each denoted as CC, CW, WC, and WW, where CC represents questions that both GPT-4o-mini and Llama3.1-70b can solve. Similarly, WW represents questions that both models fail to solve, and CW denotes questions that only GPT-4o-mini can solve. In each part, the filled bar denotes how many questions are solved by Heter-MAD, while the hollow bar denotes how many questions are not solved by Heter-MAD.

- **Do not modify the questions and only use the provided macros for your answers**.

