# OpenReview forum: "Position:  Stop Overvaluing Multi-Agent Debate—We Must Rethink Evaluation and Embrace Model Heterogeneity"
_NeurIPS.cc/2025/Position_Paper_Track — Submitted to NeurIPS 2025 Position Paper Track_

### Official Review · Reviewer_DVci · 2025-07-24

**Significance:** 4
**Presentation:** 3
**Rating:** 7
**Confidence:** 4

**Summary:**

This paper challenges the prevailing optimism around MAD frameworks for LLMs. It presents a systematic evaluation of 5 representative MAD methods across 9 widely-used benchmarks and 4 foundational LLMs. The authors find that, in their current form, MAD methods often fail to outperform simpler single-agent baselines such as CoT and SC, despite incurring greater computational cost. To address this, the paper introduces model heterogeneity, which deploy diverse LLMs within a MAD setup, to consistently improve performance. The paper advocates for rethinking MAD evaluation practices and positions model heterogeneity as a key direction for advancing the field.

**Strengths:**

1. The position is clear. The central agument is that current MAD methods are overvalued and require more rigorous evaluation. The call to embrace model heterogeneity is well-integrated into the narrative and grounded in the findings.
2. The evaluation of the paper is broad and systematic. The authors evaluate 5 methods, 9 benchmarks, and 4 foundation LLMs. Results are substantiated using detailed tables, figures, and statistical analysis, lending credibility and robustness to the argument.
3. The topic is timely and the contribution is relevant to the community. The area of MAD is rapidly growing and the evaluation of such methods is highly important.
4. The authors provide access to the code, which ensures reproducibility.

**Weaknesses:**

1. The paper misses some directly relevant citations, such as recent learning-based and uncertainty-aware MAD methods (e.g. ACC-Debate, DebUnc, etc.), which could offer promising alternatives or enhancements to the heterogeneity-focused position.
2. The paper lacks clearly articulated alternative views. While some counter-explanations (e.g. hyperparameters, task categorization, etc.) are mentioned, a dedicated section that explicitly outlines and engages with strong alternative views is lacking. For example, MAD's shortcomings may stem from immature implementations rather than conceptual flaws.

**Questions:**

1. Results (Table 1, Figure 2) show that current MAD methods often underperform CoT and SC. Could you elaborate on the underlying causes of this outcome? For example, architectural limilations or suboptimal coordination strategies?

2. How feasible is model heterogeneity in practical applications w.r.t. costs, orchestraction overhead, and inconsistencies, particularly under resource-constrained settings?

3. What is MAD's true utility (mentioned in Section 5)? Are there any concrete examples or characteristics of task domains where MAD is likely to be indispensible?

**Alternative Position:**

No

**Author Identification:**

No.

**Context:**

3

**Discussion:**

4

**Ethics:**

["NO or VERY MINOR ethics concerns only"]

**Position:**

Yes, the paper argues for or against a position related to machine learning.

**Support:**

4

**Thoroughness:**

3

---

### Official Review · Reviewer_vuxN · 2025-08-07

**Significance:** 4
**Presentation:** 3
**Rating:** 8
**Confidence:** 4

**Summary:**

This position paper argues that the field has been overvaluing multi-agent debate (MAD) methods for improving LLM performance due to fragmented evaluation practices. Through systematic evaluation of 5 MAD methods across 9 benchmarks and 4 models, the authors demonstrate that MAD methods generally fail to outperform simple single-agent baselines like Chain-of-Thought (CoT) and Self-Consistency (SC), despite consuming more computational resources. They propose that model heterogeneity, using diverse models rather than identical agents, can serve as an antidote to improve MAD performance. The paper calls for more rigorous evaluation standards and fundamental rethinking of what effective multi-agent collaboration means.

**Strengths:**

The paper's primary strength lies in its comprehensive and systematic experimental evaluation that challenges prevailing assumptions with solid empirical evidence. The scope of evaluation (5 methods × 9 benchmarks × 4 models) is impressive and provides robust statistical foundation. The introduction of model heterogeneity as a solution demonstrates constructive contribution beyond criticism. The clear articulation of future research directions provides valuable guidance for the community. The work has immediate practical implications for researchers and practitioners working with multi-agent systems.

**Weaknesses:**

* **Shallow Theoretical Analysis of MAD Underperformance:** The paper could benefit from deeper theoretical analysis of why MAD methods underperform, the current explanations remain somewhat surface-level. The authors don't explore whether the problem lies in the debate mechanisms themselves, the consensus algorithms, or the assumption that multiple identical agents can generate meaningful diversity.

* **Simplistic Model Heterogeneity:** The Heter-MAD design, randomly mixing GPT-4o-mini and LLaMA 3.1, lacks principled motivation. It ignores task-model compatibility, complementary strengths, or dynamic assignment strategies. Additionally, the paper could provide a more detailed analysis of the trade-offs involved with the proposed model heterogeneity, especially concerning the computational overhead of using multiple different LLMs.

* **Model Heterogeneity Analysis Lacks Depth:** While Table 2 shows consistent improvements from Heter-MAD, the analysis of why this works is superficial. Figure 4's categorization (CC, CW, WC, WW) is helpful but incomplete, it doesn't explain the mechanisms by which heterogeneous models achieve better consensus.

**Questions:**

* What specific benchmark characteristics do you think would better showcase MAD advantages, and how might the field develop more appropriate evaluation scenarios that require true multi-agent collaboration?
* Given your findings about model heterogeneity, how do you envision this scaling with even more diverse model pools, and what are the practical constraints for implementing this in real-world applications?

**Alternative Position:**

Yes, and alternative positions are well-considered and named but not addressed

**Author Identification:**

No.

**Context:**

3

**Discussion:**

3

**Ethics:**

["NO or VERY MINOR ethics concerns only"]

**Position:**

Yes, the paper argues for or against a position related to machine learning.

**Support:**

3

**Thoroughness:**

3

---

### Official Review · Reviewer_qWxd · 2025-08-13

**Significance:** 3
**Presentation:** 3
**Rating:** 6
**Confidence:** 4

**Summary:**

This position paper critically examines the state of multi-agent debate (MAD) research for large language models (LLMs), highlighting major gaps in current evaluation practices such as limited benchmarks, inconsistent and weak baselines, and a lack of systematic comparison among approaches. The authors conduct a comprehensive experimental study, evaluating five representative MAD methods across nine widely-used benchmarks and four foundational LLMs. Surprisingly, the results indicate that MAD frameworks often fail to outperform simple, established single-agent baselines such as Chain-of-Thought (CoT) and Self-Consistency (SC), even when consuming markedly more inference-time resources. The study further reveals that introducing model heterogeneity—i.e., using diverse foundation models as agents—leads to robust gains across frameworks, suggesting a need for the MAD community to rethink core design assumptions and evaluation methodologies.

**Strengths:**

- Comprehensive evaluation: The study systematically benchmarks five leading MAD methods (SoM, Multi-Persona, Exchange-of-Thoughts, AgentVerse, and ChatEval) on nine prominent datasets (spanning general knowledge, mathematical reasoning, and programming) using four distinct LLMs. This broad coverage, clearly presented in Table 1 (Page 5) and Figure 2 (Page 2), is a considerable strength, providing an unusually robust empirical foundation for the field.
- Valuable evaluation results: The main finding—that MAD frameworks generally do not outperform CoT/SC—directly challenges prevailing assumptions in the area. This is well-illustrated in Table 1 and further supported by the aggregated win/tie/lose statistics in Figure 2, making a compelling case for skepticism about current MAD narratives.
- Important suggestions: The discussion (Section 5) outlines not only pitfalls in the current literature but also articulates concrete, forward-looking research directions: optimizing for heterogeneity, enhancing intra-agent reasoning, and developing benchmarks revealing true utility for multi-agent collaboration.

**Weaknesses:**

- Dependence on Limited Set of LLMs: The core empirical results depend on a pool of four LLMs (e.g., GPT-4o-mini, Claude-3.5-haiku, Llama3.1:8b/70b). It is unclear how generalizable these findings are to proprietary models or to non-English or lower-resource settings. This limits the universality of the conclusions, and no discussion is provided regarding the selection criteria or potential advantages and disadvantages of the chosen models.

**Questions:**

- Besides enriching the type of tasks in the benchmark for MAD evaluation, are there any new auxiliary metrics could be suggested to identify when single agent LLM is sufficient or MAD is preferred.

**Alternative Position:**

Yes, and alternative positions are well-considered and named but not addressed

**Author Identification:**

No.

**Context:**

3

**Discussion:**

3

**Ethics:**

["NO or VERY MINOR ethics concerns only"]

**Position:**

Yes, the paper argues for or against a position related to machine learning.

**Support:**

3

**Thoroughness:**

4

---

### Note · Authors · 2025-09-01

**1-11 Submit Again:**

Probably yes

**1-1 Submission Process:**

5

**1-4 Interest:**

["Panel discussions with other position paper authors", "Structured debates on controversial topics", "Mentorship programs for early-career researchers"]

**1-5 Thoughtful:**

10

**1-6 Supportive:**

10

**1-7 Technical Aspects Versus Position:**

7

**1-9 Camera Ready Changes:**

I will include a few mentioned citations as well as

**3-1 Review Response1:**

qWxd

**3-2 Reaction To Review1:**

Thank you for your positive review on the quality of our work and strong agreement on our position. We are very encouraged that you highlighted the main strengths of our work, specifically: the comprehensiveness of our empirical evaluation, the value of our core findings in challenging current assumptions, and the importance of our suggestions for future research directions.

W.1
Our rationale for model selection was to ensure diversity and representativeness by choosing models from different model families and at different scales. Accordingly, we selected two leading proprietary models (GPT-4o-mini and Claude-3.5-haiku), as well as two widely-used open-source models (Llama3.1:8b and Llama3.1:70b) which differ substantially in scale. We will clarify these selection criteria in the revision.

While it is infeasible to be exhaustive in model and benchmark coverage, our study’s choice of 4 distinct models, 9 benchmarks, and 5 typical MAD methods represents unprecedented coverage compared to prior work (which were mostly evaluated on only 1–2 models and 3–5 benchmarks). As the reviewer noted, “This broad coverage …  is a considerable strength, providing an unusually robust empirical foundation for the field.”

Despite significant differences in the models, our key finding is remarkably consistent, which strongly suggests that the observed underperformance of MAD is not tied to any single model family or scale, but rather point to a **fundamental, model-agnostic challenge** for current MAD approaches. As the reviewer commented “our main finding … making a compelling case for skepticism about current MAD narratives.”

Q.1
As we discussed in Section 5, a promising direction for future work is to devise metrics that quantify the "knowledge diversity" required by a task. For example, a metric could assess whether a problem requires synthesizing information from multiple, disparate domains.

**3-3 Review Response2:**

vuxN

**3-4 Reaction To Review2:**

Thank you for your highly positive ranking and agreement on our position. We are particularly grateful that you recognized the paper's primary strength in its comprehensive and systematic experimental evaluation. We also appreciate your acknowledgment of our constructive contributions, such as introducing model heterogeneity, and the practical implications our work offers to the community.

Weaknesses:
We respectfully disagree with the reviewer’s criticism. As a position paper, rather than a main-track paper, this paper’s goal is to challenge the prevailing assumption about the superior performance of MAD and to encourage further exploration of model heterogeneity, stimulating greater discussion within the community. According to the CFP, position papers are NOT intended to provide exhaustive theoretical analysis or novel, well-designed methods. While we agree that MAD research would benefit from deeper theoretical analysis and methods tailored to fully leverage model heterogeneity, we believe these directions warrant dedicated work and are best addressed in future research.
Even with these listed weaknesses, the reviewer still gave strong accept ranking to this paper, which also suggests that, in the reviewer's final judgment, the paper's strengths in providing practical implications to the community outweigh the identified areas for future research.

Q.1
As stated in Section 5, we argue that scenarios naturally requiring diverse knowledge or capability from multiple agents may be suitable for MAD evaluation.

Q.2
Scaling with even more diverse models in MAD entails non-trivial challenges. As we include more agents, the growth in inference cost will be exponential, which is clearly unacceptable. A feasible solution may be selecting suitable agents and including only these agents in one round. However, this requires a good understanding of all agents’ features, capabilities, and cost, which is still challenging for LLMs nowadays.

**3-5 Review Response3:**

DVci

**3-6 Reaction To Review3:**

Thank you for your positive review and agreement on the proposed position. We are encouraged that you found our central argument to be clear and well-supported by a broad and systematic evaluation.

W.1
We thank the reviewer for referring us to those two related studies, and we will include them in our revision.

W.2
We agree with the reviewer that it is possible that “MAD's shortcomings may ...”. As shown in line 75 in our paper, “While the above results seem discouraging, we emphasize that they do not imply MAD research is a frustrating dead end.” As we further elaborated in Section 5, we contended that current methods lack "fine-grained interaction capabilities", causing agents to focus on final answers rather than scrutinizing the underlying reasoning steps. This is a critical implementation flaw in current methods, which our paper highlights as a key direction for future work. We appreciate the reviewer’s suggestion and will include these points in the alternative view section (which, as per the CFP, is not required) in our revision.

Q.1
We analyzed this in Section 3.2 and our findings, visualized in Figure 6 (Appendix D), showed that the inherent instability or conservatism, rooted in their coordination strategies, explains why they often fail to outperform simpler baselines.

Q.2
Our experiments suggest heterogeneity is practical. As shown in Table 2, combining an expensive model with the more cost-effective model consistently improved results. This indicates that performance gains do not necessarily require combining multiple expensive models; a balanced and diverse pool can be more effective.

Q.3
The MAD’s true utility, like collaboration in human society, becomes essential in scenarios that are multi-faceted and cannot be effectively solved by a single agent's knowledge/capability base.
A concrete example would be complex science analysis, which requires synthesizing knowledge from different domains and abilities in coding, analyzing, reasoning, etc.

---

### Meta-Review · Area_Chair_vwSw · 2025-09-12

**Rating:** 7
**Confidence:** 4

**Strengths:**

This paper provides a large-scale, systematic evaluation of five popular multi-agent debate (MAD) methods across nine benchmark tasks and four different language models. It finds that, contrary to common belief, these MAD methods generally do not outperform standard techniques like Chain-of-Thought or Self-Consistency. The authors present strong evidence through detailed experiments and visuals, and also propose valuable future directions such as encouraging diverse agent reasoning and better evaluation benchmarks. Their work offers both a critical reality check and constructive suggestions, making it relevant and useful for the research community.

**Weaknesses:**

Most of the weaknesses raised by the reviewers are addressed by the authors in the rebuttal.

**Questions:**

No more questions.

**Thoroughness:**

3

---

### Decision · Program_Chairs · 2025-09-26

Reject